# Associated Learning: a Methodology to Decompose End-to-End Backpropagation on CNN, RNN, and Transformer

**Dennis Y. Wu[1], Di-Nan Lin[2], Vincent F. Chen[1], Hung-Hsuan Chen[2],**

[1] Institute of Information Science, Academia Sinica

[2] Department of Computer Science, National Central University

`hibb@iis.sinica.edu.tw, lindinan934301@gmail.com`
`vincent0110@iis.sinica.edu.tw, hhchen@g.ncu.edu.tw`

## Abstract

We study associated learning (AL), an alternative methodology to end-to-end backpropagation (BP). We introduce the workflow to convert a neural network into an AL-form network such that AL can be used to learn parameters for various types of neural networks. We compare AL and BP on some of the most successful neural networks: convolutional neural networks, recurrent neural networks, and Transformers. Experimental results show that AL consistently outperforms BP on open datasets. We discuss possible reasons for AL's success, its limitations, and AL's newly discovered properties. Our implementation is available at https://github.com/Hibb-bb/AL.

## 1 Introduction

Backpropagation (BP) is the keystone of modern deep learning. Although BP is the standard way to learn network parameters, it is far from ideal. Some of the most discussed issues of BP are optimization difficulties (e.g., vanishing gradients and exploding gradients (Hochreiter et al., 2001)) and training performance (e.g., backward locking (Jaderberg et al., 2017)).

It appears that custom network structures may be needed for different types of learning tasks. Among the various forms, convolutional neural networks (CNNs), recurrent neural networks (RNNs), and Transformer networks (along with their extensions, e.g., LSTM (Hochreiter & Schmidhuber, 1997) and VGG (Simonyan & Zisserman, 2015)) are particularly useful. These networks have been successfully applied in fields as varied as computer vision, natural language processing, signal processing, and others (Goodfellow et al., 2016; Deng & Yu, 2014).

This paper studies a new learning approach—associated learning (AL)—an alternative to end-to-end backpropagation learning. AL decomposes BP's global end-to-end training strategy into several small local optimization targets such that each layer has an isolated gradient flow. However, since most layers in AL do not interact with the final loss directly, we would expect AL-training models to be less accurate than BP. Surprisingly, the original AL paper compares AL and BP using the CNN network (and its extensions, e.g., VGG) and shows impressive results based on image classification datasets (MNIST, CIFAR-10, and CIFAR-100) (Kao & Chen, 2021). We continue this line of study in two ways. First, we discover more interesting properties of AL. Second, we show how to apply AL on different network structures, including VGG (Simonyan & Zisserman, 2015), LSTM (Hochreiter & Schmidhuber, 1997), and Transformer (Vaswani et al., 2017). Eventually, we compare the networks learned via AL and via BP on various tasks (image classification, sentiment analysis, and topic classification) based on public datasets (CIFAR-10, CIFAR-100, IMDB movie reviews, AG's News, Stanford Sentiment Treebank (SST), and DBpedia). We find that AL consistently outperforms BP in most datasets. Additionally, AL requires fewer epochs than BP when using early stopping but still yields excellent accuracy. These results suggest that AL is a strong alternative to BP, as AL is effective for various tasks and various network structures.

The rest of the paper is organized as follows. In Section 2, we introduce associated learning and its properties. Section 3 presents the experiments, including a comparison of the model accuracies and

convergence speed of AL and BP on VGG, LSTM, and Transformer, a generalization test of AL and BP, and an ablation study. Section 4 reviews previous work on backpropagation alternatives and the network structures that may look similar to AL. Finally, we conclude our contribution in Section 5.

## 2 ASSOCIATED LEARNING

### 2.1 AN OVERVIEW OF AL: AL FORM, TRAINING, AND INFERENCE

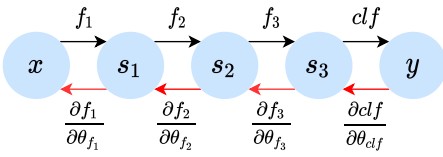

Figure 1: A classification neural network with 3 hidden layers.

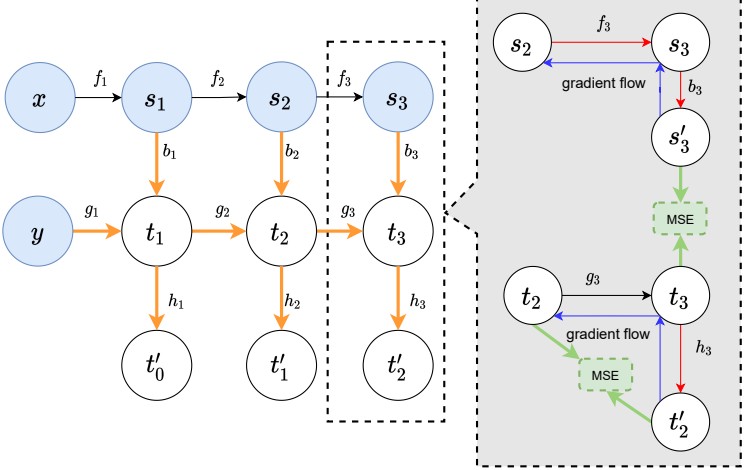

Figure 2: A 3-layer neural network in the AL form required for AL learning. Red arrows: the inference path; green arrows: the objective function; blue arrows: gradient flow; orange arrows: appended functions not in BP; black arrows: preserved functions in BP. We stop the gradient flow from an AL layer $i$ to its previous layer $i-1$ to ensure each layer is trained in isolation.

To apply AL, we must transform the network into a different structure, which we call *AL form* below. Instead of learning a function to map a feature vector $x$ to a target $y$, the AL form performs metric learning by searching for functions to transform $x$ and $y$ into latent representations such that the distance between these two latent representations is close.

Algorithm 1 shows the procedure to convert a neural work into an AL form. We use Figure 1 (a neural network with 3 hidden layers) as an example to show the process. We can regard functions $f_1, f_2, f_3$ as the encoders to convert the input feature vector into a latent representation and $clf$ as a classifier to transform the latent representation into a target. When converting this network to its AL form (referring to Figure 2), we only keep the encoders $f_i$-s and extend the model architecture by adding a bridge function $b_i$ and an autoencoder for each layer $i$. Let $s_{i-1}$ and $t_{i-1}$ be the inputs of layer $i$ (assuming $s_0 := x$ and $t_0 := y$), functions $f_i$ and $g_i$ convert $s_{i-1}$ and $t_{i-1}$ into latent representations $s_i$ and $t_i$. The function $f_i$ can be any type of forwarding block, such as a convolution layer, an LSTM layer, or a residual block in ResNet. Each bridge function $b_i$ projects $s_i$ (the latent representation of $x$ at layer $i$) to be close to $t_i$ (the latent representation of $y$ at layer $i$), i.e., $b_i(f_i(s_{i-1})) \approx g_i(t_{i-1})$. Meanwhile, $g_i$ not only extracts the latent representation of $y$, but also serves as the encoder for an autoencoder such that $h_i(g_i(t_{i-1})) \approx t_{i-1}$ ($h_i$ is the decoder). For all the AL-form networks in our experiments, each of the functions $g_i$ and $h_i$ is constructed by a linear transform followed by a non-linear activation function (e.g., tanh), and $b_i$ is a linear transformation

of the input vector or input matrix. We discuss the number of neurons for the input, output, and hidden layers of the autoencoders in Section 6.9.

To summarize, we minimize the following objective for each layer $i$ during training:

$$\mathcal{L}^i_{AL} = \mathcal{L}^i_A + \mathcal{L}^i_{AE} = \left\| b_i(f_i(s_{i-1})) - g_i(t_{i-1}) \right\|^2 + \left\| h_i(g_i(t_{i-1})) - t_{i-1} \right\|^2, \tag{3}$$

where $\mathcal{L}^i_A$ denotes the associated loss of layer $i$, and $\mathcal{L}^i_{AE}$ indicates the autoencoder loss of layer $i$.

Algorithm 2 gives the training algorithm for AL. Since each layer has its objective function (Equation 3), we can parallelly update the parameters of different layers using a pipeline to increase the training throughput (details in Section 2.2.3).

For inference, if an AL network has $L$ layers, an input $x$ goes through $f_1, \ldots, f_L$ to generate $x$'s latent representation ($s_L$), which is transformed to $y$'s latent representation ($t_L$) via $b_L$. Next, $t_L$ is converted to $y$ via $h_L, \ldots, h_1$. The autoencoders' encoding functions $g_i$-s are not used during inference. Take Figure 2 as an example, the inference path is ($f_1 \rightarrow f_2 \rightarrow f_3 \rightarrow b_3 \rightarrow h_3 \rightarrow h_2 \rightarrow h_1$). Although it could be unintuitive to concatenate two unconnected transformation functions as an inference path (i.e., $h_{i+1}$ followed by $h_i$, $i = 1, 2, 3$), it is valid because $h_{i+1}(t_{i+1}) = t'_i \approx t_i$.

---

**Algorithm 1** Converting a neural network to its AL form

    **Input** a neural network $N = (f_1, f_2, ... f_L)$, target $y \in \mathcal{R}^1$, feature vector $x \in \mathcal{R}^m$
    **Output** an AL network $A$
1:  $s_0 \leftarrow x; t_0 \leftarrow y$
2:  **for** $i = 1$ to $L$ **do**
3:      $s_i = f_i(s_{i-1})$
4:      insert a function $g_i$, where $t_i = g_i(t_{i-1})$
5:      insert a function $b_i$, where $s'_i = b_i(s_i)$                   ▷ adding bridge function
6:      insert a function $h_i$, where $t'_{i-1} = h_i(t_i)$      ▷ adding a decoder for an autoencoder
7:      $\mathcal{L}^i_{AL} = MSE(s'_i, t_i) + MSE(t_{i-1}, t'_{i-1})$     ▷ associated loss and autoencoder loss
8:  **return** $A = (a_1, \ldots, a_L), a_i = \{f_i, b_i, g_i, h_i\}^L_{i=1}$

---

**Algorithm 2** Training an AL network

    **Input** an AL network $A = (a_1, \ldots, a_L), a_i = \{f_i, b_i, g_i, h_i\}^L_{i=1}$, training features and targets $(X, Y)$
    **Output** a fine-tuned AL network $A$
1:  **repeat**
2:      Sample $x, y$ from $X, Y$.
3:      $s_0 \leftarrow x; t_0 \leftarrow y$
4:      **for** $i = 1$ to $L$ **do**
5:         $s_i \leftarrow f_i(s_{i-1})$
6:         $t_i \leftarrow g_i(t_{i-1})$
7:         $s'_i \leftarrow b_i(s_i)$
8:         $t'_{i-1} \leftarrow h_i(t_i)$
9:         $\mathcal{L}^i_A \leftarrow MSE(s'_i, t_i)$                      ▷ associated loss
10:        $\mathcal{L}^i_{AE} \leftarrow MSE(t_{i-1}, t'_{i-1})$              ▷ autoencoder loss
11:        Update the parameter of $f_i, b_i, g_i$ according to $\nabla\mathcal{L}_A$
12:        Update the parameter of $g_i, h_i$ according to $\nabla\mathcal{L}_{AE}$
13:  **until** converges

---

### 2.1.1 APPLYING AL TO DIFFERENT NEURAL NETWORK STRUCTURES

This section introduces details on converting some of the most successful neural network structures into their corresponding AL forms. We discuss vanilla CNN and VGG (used as the representative models for CNN-based models), LSTM (used as the representative model for RNN-based models), and Transformer. We also discuss how to integrate word embeddings into an AL-form network.

**CNN**    The AL form of an CNN architecture (here, vanilla CNN and VGG) uses $f_i$-s to convert an input image $x$ into latent representations by convolutions, just like a regular CNN or a VGG does. The associated loss at layer $i$ is defined as the distance between $b_i(s_i)$ (transforming the flattened feature map at layer $i$ into $t_i$'s shape) and $t_i$ (the latent representation of $y$).

**RNN**    The AL form of an RNN architecture (here, LSTM and Bi-LSTM) uses the internal state to iteratively process each element in the input sequence $x$ and generates the new internal state, just like a regular RNN does. After reading the entire input sequence $x$, we define $s_i$ the latent representation of $x$ at layer $i$ by the final internal state. Consequently, the associated loss is defined as the distance between $b_i(s_i)$ (transforming the final internal state at layer $i$ into $t_i$'s shape) and $t_i$ (the latent representation of $y$). Details are shown in Section 6.2.

**Transformer**    The AL form of a Transformer encodes the input sequence $x$ into a list of vectors, as a regular Transformer does. We define $s_i$ (the latent representation of $x$) by computing the mean-pooling on the encoded vectors. The associated loss is defined as the distance between $b_i(s_i)$ (transforming the mean-pooling output into $t_i$'s shape) and $t_i$ (the latent representations of $y$). Details are shown in Section 6.2.

**Word Embedding**    Word embeddings are frequently used as the input of LSTM or Transformer for NLP tasks. We use mean-pooling to aggregate all token's word embeddings as $s_i$.

## 2.2 Properties of AL

This section presents three properties of AL that do not exist in BP: forward shortcuts, dynamic layer accumulation, and pipelines.

### 2.2.1 Forward Shortcuts

Forward shortcuts enable faster inference. However, we can also leverage "shortcut paths" for faster inference. As shown in Algorithm 3, given an integer $\ell$ ($1 \leq \ell \leq L$), the bridge function $b_\ell$ can serve as a shortcut to transform $s_\ell$ to $s'_\ell$, which should be close to $t_\ell$ when the network is well trained. As a result, we can skip $f_j$, $b_j$, and $h_j$ for all $j > \ell$ to reduce the length of the inference function. In other words, we have multiple inference functions based on a single model. When inference time is critical, we can select a shorter inference path, e.g., $x \xrightarrow{f_1} s_1 \xrightarrow{b_1} t_1 \xrightarrow{h_1} y$. On the other hand, if inference time is unimportant, we can dynamically adjust the model complexity by modifying the number of AL layers used at the inference phase to reduce overfitting or underfitting.

### 2.2.2 Dynamic Layer Accumulation

An AL-form network allows the dynamic creation of new layers during training. Specifically, we initially create an AL-form network with $k$ AL layers and train the network based on the strategy introduced in Section 2.1. If the model still underfits, we simply add the $(k+1)$-th AL layer on top of the first $k$ AL layers. We may choose to fix the parameters in the first $k$ AL layers and train only the parameters in the $(k+1)$-th AL layer. In contrast, when using a standard neural network, adding layers dynamically is more complicated.

### 2.2.3 Pipeline

In an AL-form network, the network parameters in different AL layers can be trained simultaneously via pipelines because each AL layer $i$ has a local objective function. In contrast, it is difficult to simultaneously update the parameters of different layers for a standard BP-based neural network.

Given $n$ training instances in a standard neural network with $L$ hidden layers, if we apply BP for training, the time complexity is $O(nL)$ because each instance must complete the entire forward ($L+1$ transformations) and backward ($L+1$ transformations) process. Thus, for such an architecture, the time complexity increases linearly with the total number of layers $L$. However, since each AL layer has its local objective, each layer can update its parameters without waiting for the gradient from preceding layers. As a result, for an AL-form network with $L$ AL layers, using $L$ computation units can make the parameter update process fully pipelined, as explained below.

We denote $task_i^j$ as the task of computing both forward and backward propagation for an AL layer $i$ based on the $j$-th instance. If an AL layer $i$ finishes $task_i^1$ at time step $t$, at the next time step $t+1$, AL layers $i$ can continue performing $task_i^2$ since all parameters in this layer are updated based on $task_i^1$. Meanwhile, AL layer $i+1$ performs $task_{i+1}^1$. Consequently, training based on the first instance requires $O(L)$, and each of the following $n-1$ instances requires extra $O(1)$ time units. The time complexity of one training epoch eventually reduces to $O(n+L)$.

## 3 Experiments

To evaluate how AL works on different network architectures, we conducted experiments on image classification with vanilla CNN and VGG and on sentiment and topic classification with LSTM and Transformer (using pre-trained word embeddings as input). In addition, we conducted a case study on AL's generalization ability, evaluated by fitting AL models and BP models on different portions of noisy labels. Finally, we discuss possible reasons for AL's success based on ablation studies. We report more detailed settings in the Appendix.

### 3.1 Datasets

We used four datasets (IMDB Review, AG's News corpus, DBpedia Ontology, and the Stanford Sentiment Treebank) to compare AL and BP on LSTM and Transformer, and two datasets (CIFAR-10 and Fashion-MNIST) to compare AL and BP on CNN and VGG. Dataset details are in Appendix 6.4

### 3.2 Text Classification (LSTM and Transformer)

We compared BP and AL on LSTM and Transformer based on the text classification task.

For LSTM, we used a 2-layer bidirectional LSTM with pre-trained GloVe word embeddings (Pennington et al., 2014) as input. For Transformer, we used a 2-layer Transformer encoder, again with pre-trained GloVe word embeddings. During training, we recorded the accuracies of the validation set for every epoch and saved that model with the best performance on the validation set for testing. For each model, we repeated the process five times and reported the average accuracy and standard deviation. For a fair comparison, we forced the AL parameter counts to not exceed those of the compared standard neural network by reducing AL's hidden dimension size.

Table 1: Accuracy (mean $\pm$ standard deviation) of neural networks and corresponding AL versions on text classification datasets. Tran denotes Transformer, <X>-AL-full denotes full path inference on AL-form network <X>, and <X>-AL-SC$i$ denotes shortcut inference through bridge function $b_{i+1}$. EMB-AL-SC indicates inference through the shortcut in the embedding layer, i.e., after computing mean-pooling on the word embeddings, the output is passed to a bridge function.

| Method | IMDB | AGNews | SST | DBpedia |
|---|---|---|---|---|
| LSTM | $86.25 \pm 0.63$ | $90.32 \pm 0.23$ | $\mathbf{82.23 \pm 0.93}$ | $97.34 \pm 0.11$ |
| LSTM-AL-full | $86.41 \pm 0.61$ | $\mathbf{91.53 \pm 0.20}$ | $81.33 \pm 0.31$ | $\mathbf{98.30 \pm 0.01}$ |
| LSTM-AL-SC1 | $86.16 \pm 0.22$ | $91.42 \pm 0.17$ | $80.83 \pm 0.42$ | $98.21 \pm 0.06$ |
| EMB-AL-SC | $\mathbf{87.80 \pm 0.13}$ | $91.03 \pm 0.28$ | $80.57 \pm 0.81$ | $96.95 \pm 0.03$ |
| Tran | $83.45 \pm .033$ | $90.71 \pm 0.28$ | $76.48 \pm 0.55$ | $97.37 \pm 0.06$ |
| Tran-AL-full | $85.65 \pm 0.77$ | $91.17 \pm 0.43$ | $\mathbf{81.16 \pm 0.11}$ | $\mathbf{97.55 \pm 0.06}$ |
| Tran-AL-SC1 | $86.25 \pm 1.34$ | $\mathbf{91.49 \pm 0.09}$ | $80.58 \pm 0.67$ | $96.53 \pm 0.29$ |
| EMB-AL-SC | $\mathbf{86.98 \pm 0.72}$ | $91.43 \pm 0.11$ | $80.12 \pm 0.57$ | $97.14 \pm 0.11$ |

We report the accuracies of LSTM and Transformer trained via BP and AL in Table 1. We also report the accuracies based on the predictions using shorter inference paths in the same Table. Although AL is trained by isolated gradient flows, AL-form LSTMs outperform their standard BP-trained counterparts on most datasets. Transformer-AL also outperforms Transformer on every dataset. Also, we observe that standard Transformer easily overfits smaller datasets like SST and IMDB, resulting in poor test accuracies, but Transformer-AL does not seem to have this weakness.

In Table 2 and Table 3, we report the epoch that yielded the best results. We discover that LSTM-AL converges much faster than BP, suggesting that training time will be improved if applying early

Table 2: Accuracy (mean $\pm$ standard deviation) and epoch (mean $\pm$ standard deviation) to reach the best accuracy using different RNN models and their AL versions on IMDB and AGNews.

| Method | IMDB | | AGNews | |
|---|---|---|---|---|
| | Epoch | Accuracy | Epoch | Accuracy |
| LSTM | $15.4 \pm 2.60$ | $86.25 \pm 0.63$ | $25.4 \pm 13.42$ | $90.32 \pm 0.23$ |
| LSTM-AL | $\mathbf{4 \pm 1.22}$ | $\mathbf{86.41 \pm 0.61}$ | $\mathbf{2.4 \pm 0.89}$ | $\mathbf{91.53 \pm 0.20}$ |
| Tran | $35 \pm 6.63$ | $83.45 \pm 0.33$ | $\mathbf{3.75 \pm 0.96}$ | $90.71 \pm 0.28$ |
| Tran-AL | $\mathbf{4.4 \pm 0.55}$ | $\mathbf{85.65 \pm 0.77}$ | $20.25 \pm 12.39$ | $\mathbf{91.17 \pm 0.43}$ |

Table 3: Accuracy (mean $\pm$ standard deviation) and epoch (mean $\pm$ standard deviation) to reach the best accuracy using different RNN models and their AL versions on SST and DBPedia.

| Method | SST | | DBpedia | |
|---|---|---|---|---|
| | Epoch | Accuracy | Epoch | Accuracy |
| LSTM | $16 \pm 2.65$ | $\mathbf{82.33 \pm 0.93}$ | $31.8 \pm 5.26$ | $97.34 \pm 0.11$ |
| LSTM-AL | $\mathbf{2.25 \pm 1.89}$ | $81.33 \pm 0.31$ | $\mathbf{14.6 \pm 2.97}$ | $\mathbf{98.30 \pm 0.01}$ |
| Tran | $\mathbf{6 \pm 2.1}$ | $76.48 \pm 0.55$ | $\mathbf{6.75 \pm 1.71}$ | $97.37 \pm 0.06$ |
| Tran-AL | $9.67 \pm 0.58$ | $\mathbf{81.16 \pm 0.11}$ | $13.75 \pm 2.22$ | $\mathbf{97.55 \pm 0.06}$ |

stopping. For the Transformer-AL, although the best validation and test accuracy come at roughly the 20th epoch on AGNews, Transformer-AL reaches high validation and test accuracy within the first 5 epochs. Since we report only the epoch that reaches the best validation and test accuracies, the required epochs look large. Particularly, Transformer-AL achieved $99\%$ of the best validation set accuracy before the 5th epoch. More details can be found in Appendix 6.7.

## 3.3 IMAGE CLASSIFICATION (VANILLA CNN AND VGG)

We also reproduced the experiments in Kao & Chen (2021) to determine whether AL-form CNNs also converge faster. In image classification, we report performance on the CIFAR-10 and Fashion-MNIST benchmarks. We selected FashionMNIST instead of MNIST because MNIST is likely too simple to show the power of different models: CIFAR-10 is more challenging than FashionMNIST since it contains three channels, whereas FashionMNIST is limited to grayscale images. Table 4 shows the results: all AL models converge faster than BP. Figure 6 in Appendix 6.6 shows the relationship between epochs and test accuracies on CNN and CNN-AL. Also, AL models perform slightly better in terms of accuracy, consistent with the results demonstrated in Kao & Chen (2021).

Table 4: Accuracy (mean $\pm$ standard deviation) and epoch (mean $\pm$ standard deviation) to reach the best accuracy using different CNN models and their AL versions.

| | CIFAR-10 | | Fashion MNIST | |
|---|---|---|---|---|
| | Epoch | Accuracy | Epoch | Accuracy |
| CNN | $36.8 \pm 4.60$ | $82.64 \pm 0.61$ | $58.4 \pm 9.32$ | $91.57 \pm 0.11$ |
| CNN-AL | $\mathbf{33.6 \pm 4.97}$ | $\mathbf{85.16 \pm 0.11}$ | $\mathbf{11.6 \pm 1.67}$ | $\mathbf{92.07 \pm 0.18}$ |
| VGG | $47.2 \pm 7.82$ | $92.14 \pm 0.17$ | $32 \pm 4.34$ | $\mathbf{94.18 \pm 0.04}$ |
| VGG-AL | $\mathbf{35.6 \pm 2.19}$ | $\mathbf{92.48 \pm 0.15}$ | $\mathbf{25 \pm 3.83}$ | $93.80 \pm 0.08$ |

## 3.4 GENERALIZATION

To assess the generalizability of a model, a common approach is to empirically compare the errors on the training data and test data. Garg et al. (2021) show another way to evaluate generalizability: a model is prone to overfitting if it closely fits a dataset with random labels. We use both methods to show that an AL-trained model may be more general than that trained by BP.

We took $\theta$ proportion of data, in which we changed the labels randomly. We used different proportions to train models. We report both the training and test accuracies on clean test data.

Table 5 shows the results. First, as shown in the last column (for a label noise rate of $\theta = 0$), the training accuracy of the CNN is larger than that of the CNN-AL, but CNN-AL outperforms CNN

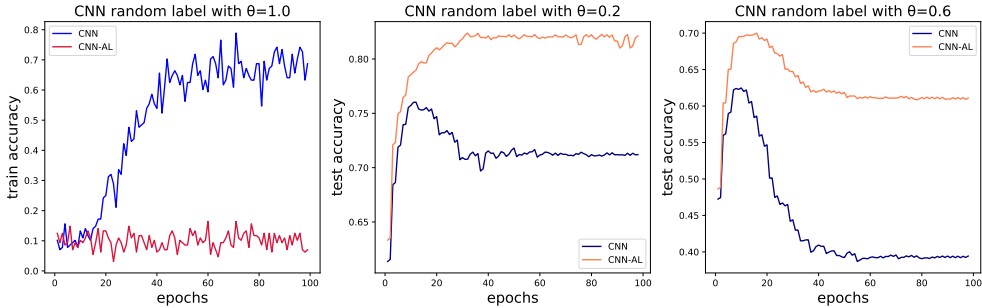

Figure 3: Training accuracy when $\theta = 1.0$ (left) and test accuracies when $\theta$ equals 0.2 (middle) and 0.6 (right) with different epochs.

in terms of test accuracy. Second, with $\theta > 0$, the vanilla CNN network with BP training fits the randomly labeled training data much better than the AL-form CNN. However, if we evaluate model accuracies based on the clean test data, the AL-form CNN performs better. Taken together, the above empirical evidence suggests that the new model may generalize better (Garg et al., 2021). The relationship between accuracy and training epochs (for $\theta = 1.0, 0.2,$ and 0.6) is given in Figure 3.

Table 5: Accuracies of different label noise rates.

|  | $\theta$ | 1.0 | 0.8 | 0.6 | 0.4 | 0.3 | 0.2 | 0.1 | 0.0 |
|---|---|---|---|---|---|---|---|---|---|
| Training | CNN | 68.75 | 64.84 | 82.03 | 93.75 | 89.84 | 96.88 | 96.09 | 97.48 |
|  | CNN-AL | 8.59 | 41.41 | 50.00 | 65.63 | 74.22 | 78.13 | 85.94 | 96.26 |
| Testing | CNN | 9.96 | 23.25 | 39.23 | 56.30 | 63.60 | 71.18 | 77.04 | 82.64 |
|  | CNN-AL | 10.00 | 38.23 | 64.74 | 76.77 | 78.70 | 82.19 | 83.40 | 85.16 |

## 3.5 WHAT MAKES AL WORK – AN ABLATION STUDY

We empirically validates that AL generates great models, converges fast, and generalizes well. Below we report ablation studies using AG's news dataset to understand why AL performs excellently.

First, we ensure the gradient flow in each AL layer does not influence the gradients in other layers via the gradient stopping mechanism.[1] However, to optimize the objective function (Equation 3), either allowing or blocking the gradient flow between AL layers is acceptable. We sought to determine whether allowing gradient flow between AL layers improves model accuracy (although such a design may make pipelining infeasible). Hence, we disabled all the gradient stops (allowing gradients to flow through different AL layers). The results are denoted as "LSTM-AL w/o grad. stops" in Table 6: accuracy is nearly unchanged. Therefore, given enough computing nodes, applying a pipelined AL increases training throughput without sacrificing model accuracy.

Second, to test the effect of associated loss, we disabled the gradient stops and removed the associated loss (except the last AL layer) from the model. The second row of Table 6 shows the result: although the accuracy is similar to that with associated loss, far more epochs are needed to converge. So, the bridge functions likely facilitate model training by gradually bringing $s_i'$ and $t_i$ closer.

With the third experiment, we sought to investigate the effect of the autoencoder components in the AL model. To do this, we passed the output of a normal Bi-LSTM to a bridge function, whose output $t_1$ was fed to the bottleneck layer of an autoencoder, which decodes $t_1$ back to the target $y$. Compared with the original Bi-LSTM classifier, the new design results in higher accuracy, as seen in the third (LSTM + AE*1) in Table 6. This suggests that replacing the standard classifier with a decoder of an AE and training it with AL objectives could produce a better model. Additionally, we experimented with removing all autoencoders and using $b_i$ as the classifier to predict $y$ for each AL layer (similar to the early exit technique in Teerapittayanon et al. (2016)). We use "LSTM-AL w/o AE (SC-1)" and "LSTM-AL w/o AE (full)" in Table 6 to denote "early exit at the first layer" and "using $b_L$ the last bridge as the classifier", respectively. The results show that using autoencoders

---

[1]This is supported by `Tensor.detach()` in PyTorch and `tf.stop_gradient()` in TensorFlow.

gives a better predicting model. We suspect that each autoencoder perhaps performs some kind of feature extraction and regularization. Particularly, although it is easy to transform one-hot encoded $y$ to $t_1$ to decrease associated loss and transform $t_1$ to $t_0'$ to decrease the autoencoder loss, the network still needs to learn more challenging tasks: converting $t_i$ to $t_{i+1}$ and $t_{i+1}$ to $t_i'$ for $i > 1$ to reduce the associated loss and autoencoder loss of layer $i$. Consequently, adding autoencoders may prevent overfitting for the components beyond layer 1.

Finally, for LSTM-AL, the inference path goes through 4 transformation functions ($b_3 \rightarrow h_3 \rightarrow h_2 \rightarrow h_1$) after the LSTM module. To ensure a fair comparison, we also tested a network with a 3-layer MLP classifier after the standard LSTM model. The result is denoted as LSTM + MLP*3 in Table 6. As shown, replacing the MLP from one layer to three layers does not produce a better model, i.e., the number of layers is not why LSTM-AL is a better model.

Table 6: Ablation studies on AG's News dataset to study why AL works.

| Method | Accuracy | Convergence epochs |
|---|---|---|
| LSTM-AL w/o grad. stops | $91.39 \pm 0.24$ | $2.2 \pm 0.45$ |
| LSTM-AL w/o $\mathcal{L}_A^i$ ($i = 1, \dots, L-1$) and w/o grad. stops | $91.48 \pm 0.15$ | $23.8 \pm 6.83$ |
| LSTM + AE*1 | $90.75 \pm 0.13$ | $27.5 \pm 7.57$ |
| LSTM w/o AE (SC-1) | $89.30 \pm 0.88$ | $7.75 \pm 7.51$ |
| LSTM w/o AE (full) | $89.39 \pm 0.99$ | $7.75 \pm 7.51$ |
| LSTM + MLP*3 | $90.15 \pm 0.15$ | $37.6 \pm 2.89$ |
| LSTM | $90.32 \pm 0.23$ | $25.4 \pm 13.42$ |
| LSTM-AL | $91.53 \pm 0.20$ | $2.4 \pm 0.89$ |

## 4 RELATED WORK

### 4.1 BACKPROPAGATION ALTERNATIVES

Backpropagation is fundamental in deep learning. However, backpropagation suffers from optimization and performance issues. Related work usually concerns methodologies that better imitate the signal transmission process of biological neural networks, since biological neural networks are highly efficient in analyzing visual, audio, textual, and other signals (Lee et al., 2015; Kao & Chen, 2021; Nøkland, 2016; Ororbia et al., 2018; Ororbia & Mali, 2019). Alternatives to end-to-end backpropagation can be categorized into three types: proxy objective, target propagation, and synthetic gradients (Duan & Principe, 2021). Below we introduce each kind and the representative works.

The first type—proxy objective—uses a local objective function for one layer $i$ to learn $\boldsymbol{\theta}_i$, the parameters for the current layer $i$. The model then fixes $\boldsymbol{\theta}_i$, adds one more layer $i+1$ with another objective function, and learns $\boldsymbol{\theta}_{i+1}$, the parameters for the new layer. Consequently, proxy objective decouples the end-to-end backpropagation since the parameters in each layer are trained via a local objective function (Nøkland & Eidnes, 2019; Belilovsky et al., 2019; 2020; Löwe et al., 2019).

The second type—target propagation—approximates $g_i$, the inverse of the downstream layers for every layer $i$. To update $\boldsymbol{\theta}_i$, the parameters of layer $i$, the model backpropagates the target (not gradient) through function $g_i$. Consequently, target propagation handles the situation where the relationship between the parameters and the cost is highly non-linear (e.g., discrete), which is otherwise difficult to solve using backpropagation (Lee et al., 2015; Meulemans et al., 2020).

The third type is synthetic gradients, which learns a function to approximate the gradient for each layer and utilizes the approximated gradients as the true gradients to update the parameters. This method thus decouples the layer-wise parameters (Jaderberg et al., 2017; Czarnecki et al., 2017; Lansdell et al., 2019).

Associated learning is different from these methods in several ways. First, both proxy gradient and target propagation propagate the signal (loss or target) from the output layer directly to each layer. Biologically, this could be implausible because when in deep networks, hidden layers near the inputs are unlikely to receive direct signals from the output layer that are far away. Associated learning, on the other hand, does not require each component to receive signals directly from the output layer. Additionally, although proxy gradient allows each layer to have a local objective, most models of

this type still learn the parameters in a layer-wise fashion, so it may still be challenging to learn the parameters using a pipeline. One exception is Greedy InfoMax (GIM) (Löwe et al., 2019), which does not require a direct signal from the output layer. However, GIM uses contrastive loss as the loss function, so GIM can mainly be applied to self-supervised learning tasks. As a result, for supervised learning tasks, AL is likely a more natural choice than GIM. As for synthetic gradients, empirical experiments show that gradients are difficult to predict, so model training based on synthetic gradients usually yields much worse accuracies than those for models trained via backpropagation.

## 4.2 Neural Networks with Structures Similar to AL

The structure of an AL-form model may look similar to a ResNet, as the bridge functions in AL appear analogous to the residual connections in ResNet. Therefore, some may argue that AL's superiority is the result of its resemblance to ResNet. However, the residual connections in ResNet are implemented to mitigate accuracy saturation (He et al., 2016), and all the parameters are adjusted to fit a global objective function, whereas the bridge functions in AL are implemented to modularize the network such that the gradient flow in one AL layer does not pass to other layers. Therefore, the bridge functions and the residual connections are only structurally similar; their purposes and functionalities are entirely different.

An AL-form network may also appear similar to contrastive learning (CL). Particularly, CL transforms a pair of images $i$ and $j$ (along with their augmented images $i'$ and $j'$) into vectors with large distances between (1) images $i$ and $j$ and (2) augmented images $i'$ and $j'$, with small distances between (1) an image $i$ and its augmented image $i'$ and (2) an image $j$ and its augmented image $j'$. This process is related to AL because both models convert inputs to vectors and compare the distance between the transformed vectors. Following this line, we are interested in studying the relationship between AL and CL in two ways. First, we could apply AL to CL such that CL is trained more effectively. Second, we could extend the AL model such that the inputs include a pair of samples $x_1$ and $x_2$ with different labels and design an objective function that maximizes the distance between the vectors transformed from $x_1$ and $x_2$ (since their labels are different) and simultaneously minimizes the current AL loss function (Equation 3).

An AL network is also structurally similar to the models in (Lin et al., 2014; Yeh et al., 2017), which also transform $x$ into a latent space and convert it to $y$ via an autoencoder. However, the design motivations are very different from AL. The two works attempt to efficiently represent a multi-label target $y$ by a low-dimensional latent representation $c$, so learning a function to convert $x$ to $c$ may be easier. The motivation of AL is to design isolated loss functions so that the new design has favorable properties that BP does not have, such as short gradient flows, forward shortcuts, dynamic layer accumulation, and layer-wise pipeline learning. Isolated objectives are not implemented in (Lin et al., 2014; Yeh et al., 2017).

Ladder network is another structurally similar network (Rasmus et al., 2015), which adds skip connections between each encoder to its corresponding decoder. While the ladder network may look similar to AL structurally, at least two manifest differences exist. First, the residual between $y$ and $\hat{y}$ in the ladder network is propagated through all layers to update the parameters on the inference path, but in AL, each layer has a local objective function, and most layers do not receive signals from the output layer. Second, the skip connection in the ladder network connects an encoder layer to a decoder layer. In contrast, the bridge function in AL connects a hidden representation of $x$ to the bottleneck layer of an autoencoder to create the local loss.

## 5 Conclusion

This paper studies associated learning, an alternative methodology to end-to-end backpropagation. Since backpropagation is the cornerstone of today's deep learning but still suffers from optimization and training performance issues, it is appropriate to study alternatives to backpropagation. We discussed the unique properties of AL that BP does not have. AL provides multiple inference paths; it allows dynamic layer accumulation during training; it supports pipeline training, and, perhaps most importantly, its prediction power is comparable to and frequently better than backpropagation on many of the most successful neural network structures. Ablation studies disclose why AL may work and demonstrate AL's effectiveness and architecture flexibility in training and testing.

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

## 6 APPENDIX

### 6.1 AL INFERENCE ALGORITHM

Algorithm 3 shows the inference algorithm for AL networks. It supports forward shortcuts by going through only $\ell$ of the $L$ AL layers.

From Algorithm 3, it is clear that the functions $g_i$-s are not used in inference, and only one bridge function out of the $L$ $b_i$-s are used in inference. Therefore, the number of parameters of inference is smaller than the number of training parameters.

---

**Algorithm 3** Using AL form network for inference

---

**Input** AL network $A = (a_1, a_2, ...a_L)$, $a_i = \{f_i, b_i, g_i, h_i\}_{i=1}^L$, input data $x$, inference length $\ell, 1 \leq \ell \leq L$
    **Output** predicted label
1:  $s \leftarrow x$
2:  **for** $i = 1$ to $\ell$ **do**
3:      $s \leftarrow f_i(s)$
4:  $t \leftarrow b_\ell(s)$
5:  $t' \leftarrow t$                                               ▷ Same as Figure 2
6:  **for** $i = \ell$ to $1$ **do**
7:      $t' \leftarrow h_i(t')$
8:  **return** $t'$

---

### 6.2 CNN, RNN, AND TRANSFORMER IN THEIR AL FORMS

Here we extend 2.1.1 for further explanation. In AL, the process in bridge layers did not influence the actual forward path in AL. Therefore, the process in 2.1.1 is only used to calculate associated loss and will not effect the original representation shape. Figure 4 and Figure 5 show that when we take the last hidden state in the sequence input to calculate associated loss, the original sequence representation remains the same when propagate to the next layer.

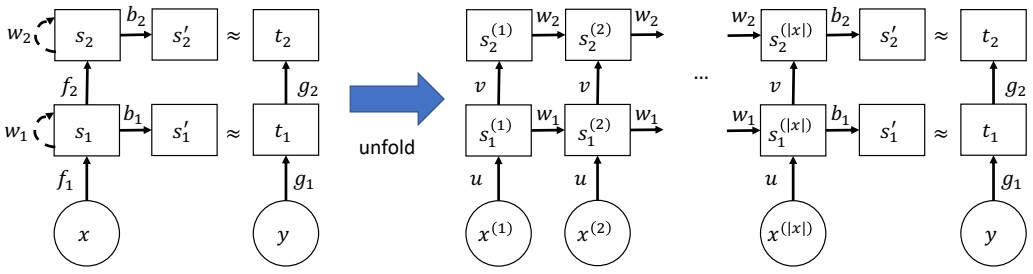

Figure 4: A closer look on how AL works on RNN.

### 6.3 IMPLEMENTATION ENVIRONMENT DETAILS

For LSTM and Transformer, we used the PyTorch package (Paszke et al., 2019). For CNN, we used the Tensorflow Abadi et al. (2015) package to build the AL models (vanilla CNN and a visual geometry group (VGG) network). We trained the models with GeForce 3090 GPUs on Ubuntu Linux 20.04. Since current hardware development does not support actual pipeline training, we simulated pipeline training by blocking gradient flows between AL layers with `Tensor.detach()` in PyTorch and with `tf.stop_gradient()` in TensorFlow.

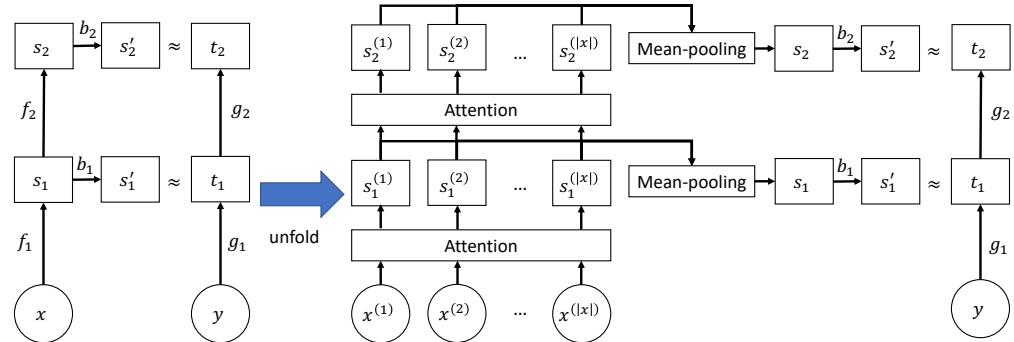

Figure 5: A closer look on how AL works on Transformer.

## 6.4 DATASET STATISTICS

The IMDB Review dataset (Maas et al., 2011) is a binary sentiment classification dataset containing 25k training data and 25k test data. AG's News corpus dataset (Zhang et al., 2015) includes four news topics: world, sports, business, and sci/tech. It has 120k training data and 7.6k test data. The DBpedia Ontology classification dataset (Zhang et al., 2016) was constructed by picking 14 non-overlapping topic classes from DBpedia 2014, containing 560k training samples and 70k test samples. The Stanford Sentiment Treebank (SST2) dataset (Socher et al., 2013) is a binary sentiment classification dataset from GLUE (Wang et al., 2018). The dataset contains 67k training samples, 872 validation samples, and 1.8k test samples. The CIFAR-10 dataset contains color images from 10 classes; 50k are training samples, and 10k are test samples. Finally, the Fashion-MNIST dataset (Xiao et al., 2017) is a grayscale image classification dataset with 10 classes. It consists of 60k training samples and 10k test samples.

## 6.5 TEXT CLASSIFICATION

During training, the most frequent 30k words in the training corpus were included in the word embedding. We replaced the remaining words with the [UNK] token. We trained all text classification experiments with 50 epochs. We used the Adam optimizer.

### 6.5.1 BI-LSTM

For Bi-LSTM, we applied gradient clipping with the value 5. For the standard version of Bi-LSTM, we experimented with learning rates of 0.0005 and 0.0001 and reported that with better accuracy. We also adjusted the learning rate for AL, but the learning rate has little influence on the accuracy and the convergence speed. We used a batch size of 64 for all Bi-LSTM experiments. For IMDB and DBpedia, we set the max sequence length to 500. For AG's News and SST, we set the max sequence length to the longest sequence length; the exact length settings were applied to the Transformer experiments. The objective of the standard Bi-LSTM was a cross-entropy loss. Details are given in Table 7.

Table 7: Bi-LSTM settings.

|  | emb dim | lstm1 dim | lstm2 dim | $g$ dim | $h$ dim | lr |
|---|---|---|---|---|---|---|
| LSTM | 300 | 350 | 350 | – | – | 0.0001 |
| LSTM-AL | 300 | 300 | 300 | 128 | 128 | 0.0001 |

### 6.5.2 TRANSFORMER

For the standard Transformer model, we used negative log-likelihood as the objective with multi-head attention (6 heads). We set the batch size to 128 because the standard Transformer converges only with a large batch size in our experiment. Table 8 gives the details.

Table 8: Transformer settings.

|          | emb dim | layer1 dim | layer2 dim | head | $g$ dim | $h$ dim | lr |
|----------|---------|------------|------------|------|---------|---------|---------|
| Tran     | 300     | 512        | 512        | 6    | –       | –       | 0.00025 |
| Tran-AL  | 300     | 256        | 256        | 6    | 128     | 128     | 0.00025 |

### 6.6 IMAGE CLASSIFICATION

We normalized the image pixels for both FashionMNIST and CIFAR-10 datasets as a preprocessing step. Since CIFAR-10 is more challenging, we also performed data augmentation on this dataset by resizing, random cropping, random flipping, and adjusting the brightness of the images. We set the batch size to 128. We set the initial learning rate to 0.0001 and applied scheduled decay until converging. We used Adam as the optimizer and cross-entropy loss for the standard version.

**Vanilla CNN Model**  We implemented the vanilla CNN by modifying the previous code. The CNN contains 13 hidden layers and an output layer. The first 8 hidden layers are convolution layers; the last is flattened to a fully connected layer with 1280 neurons. The subsequent layers consist of 4 feedforward layers, followed by an output layer with 10 classes. We set the initial learning rate to 0.0001 and applied scheduled decay.

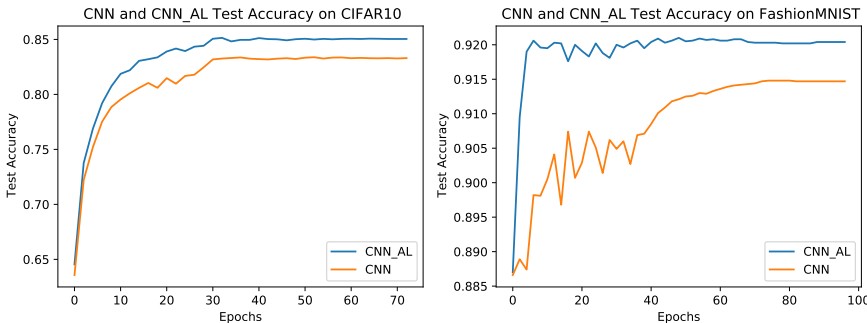

Figure 6: Testing accuracy vs. epochs for CNN and CNN-AL on CIFAR-10 and FashionMNIST.

### 6.7 TRANSFORMER-AL CONVERGENCE SPEED

Training an LSTM, a CNN, or a VGG based on AL requires much fewer epochs when compared with training them by BP; this trend is less clear when training a Transformer-AL model. However, we found that after training the Transformer-AL for roughly 5 epochs, the validation set accuracy (90.95) can already achieve 99% of the best validation set accuracy (91.31), and the test accuracy (90.98) is very close to the best test accuracy (91.17, i.e., training Transformer-AL for 20 epochs for AG News). So, we could reduce the number of epochs by sacrificing slight accuracy. Transformer-AL on DBpedia has a similar trend: when training Transformer-AL on DBpedia for 5 epochs, its validation set accuracy achieves 96.0902, which is very close to the best validation set accuracy (96.6134). This result shows that Transformer-AL is efficient on convergence when training on DBpedia and AGnews.

## 6.8 Associated Loss vs. Layer Depth

This section shows that, under the AL network structure, the latent representations in the lower layer appear to help generate the latent representations in an upper layer.

We experimented with the AL layer number $i$ vs. the associated loss $\|s_i - t_i\|_2^2$ using different datasets on different network structures. The following tables show that the associated loss at an upper layer (i.e., the layers closer to the output) is smaller than at a lower layer (i.e., the layers closer to the input). Consequently, the shortcuts are likely doing some kind of curriculum learning.

| Layer | Associated Loss on DBPedia | Associated Loss on AGNews |
|---|---|---|
| Embedding layer | $5.6061 \times 10^{-5}$ | $1.4802 \times 10^{-4}$ |
| LSTM-1 | $3.7440 \times 10^{-5}$ | $5.3281 \times 10^{-5}$ |
| LSTM-2 | $1.7786 \times 10^{-7}$ | $4.8651 \times 10^{-6}$ |

Table 9: Layer depth vs. associated loss on LSTM-AL

| Layer | Associated Loss on DBPedia | Associated Loss on AGNews |
|---|---|---|
| Embedding layer | $5.8318 \times 10^{-5}$ | $3.3179 \times 10^{-4}$ |
| Layer-1 | $3.0053 \times 10^{-6}$ | $1.2561 \times 10^{-4}$ |
| Layer-2 | $7.4727 \times 10^{-8}$ | $1.3976 \times 10^{-5}$ |

Table 10: Layer depth vs. associated loss on Transformer-AL

## 6.9 The Design of the Overcomplete Autoencoders in AL

An autoencoder's hidden layer usually consists of fewer neurons than the input and output layers. However, we use an overcomplete autoencoder (i.e., the hidden layer has more neurons than the input/output layers) (Vincent et al., 2010) in AL. An overcomplete autoencoder can reach zero reconstruction loss (i.e., the autoencoder loss in Equation 3) by copying the neurons from the input layer to the output layer. However, in AL, $g_i$ (the encoding function of the autoencoder at layer $i$) needs to minimize both $\mathcal{L}_{AE}^i$ the autoencoder loss and $\mathcal{L}_A^i$ the associated loss. As a result, $\mathcal{L}_A^i$ can be regarded as a regularization term when minimizing $\mathcal{L}_{AE}^i$.

Additionally, we hypothesize that mapping the one-hot encoded $y$ to a high dimensional vector can accelerate the convergence speed. According to Graf et al. (2021), the cross-entropy loss and supervised contrastive loss attain their minimum if and only if the features of each class collapse to the vertices of an origin-centered regular $K - 1$ simplex, where $K$ is the number of classes. In AL, since the autoencoder in layer-1 simply maps $y$ to a high dimensional space and back (i.e., letting $h_1(g_1(y)) \approx y$), it is nearly guaranteed to reach minimum loss. Therefore, we can consider $g_1(y)$ as an ideal prototype for AL to fit in, such that, $b_1(f_1(x)) \approx g_1(y)$. Therefore, the associated loss directly optimizes for a $K - 1$ regular simplex in the representation space, which is the best configuration to minimize the supervised contrastive learning loss (Khosla et al., 2020). Such an autoencoder setup probably helps AL find the best configuration in the early stage of training. This may explain why AL converges so fast. Besides, an overparameterized model may accelerate optimization (Arora et al., 2018; Chen & Chen, 2020). Empirical studies in Section 3.5 support our claim: using overcomplete autoencoders to replace classifiers performs better.

## 6.10 PyTorch Pseudocode

Here we provide a PyTorch pseudocode of a 5-layer AL network.

```python
import torch
import torch.nn as nn

class MLP(nn.Module):
    def __init__(self, d1, d2):
        super().__init__()
        self.f = nn.Sequential(
            nn.Linear(d1,d2),
            nn.Sigmoid())
    def forward(self, x):
        return self.f(x)

class AL(nn.Module):
    def __init__(self, class_num):
        super().__init__()
        # we choose 10 as hidden dimension size for simplicity.
        self.f = nn.ModuleList([MLP(10, 10) for i in range(5)])
        self.g = nn.ModuleList([MLP(10, 10) for i in range(5)])
        self.b = nn.ModuleList([MLP(10, 10) for i in range(5)])
        self.h = nn.ModuleList([MLP(10, 10) for i in range(5)])
        self.g[0], self.h[0] = MLP(class_num, 10), MLP(10, class_num)

        self.mse, self.ce = nn.MSELoss(), nn.CrossEntropyLoss()
        self.c = class_num

    def forward(self, x, y):

        loss = 0.0
        s_i, t_i = x, nn.functional.one_hot(y, num_classes=self.c).float()
        # convert y into one-hot vector for linear input
        for i in range(5):

            last_t_i, s_i = t_i, self.f[i](s_i) # encode input
            t_i, s_i_prime = self.b[i](s_i), self.g[i](t_i) # bridge + encode target
            t_i_prime = self.h[i](t_i) # decode target

            if i == 0:
                loss += self.mse(t_i.detach(), s_i_prime) + self.ce(last_t_i, t_i_pr:
            else:
                loss += self.mse(t_i.detach(), s_i_prime) +
                self.mse(last_t_i, t_i_prime)
            s_i, t_i = s_i.detach(), t_i.detach() # cut gradient

        return loss

    def inference(self, x):
        for layer in self.f:
            x = layer(x)
        y = self.b[-1](x)
        for layer in reversed(self.h):
            y = layer(y)
        return y

model = AL(class_num=5)
```

```
x, y = torch.rand(30, 10, dtype=torch.float), torch.randint(high=4, size=(30, ))
loss, pred = model(x, y), model.inference(x)
```

