# OpenReview forum: "Associated Learning: an Alternative to End-to-End Backpropagation that Works on CNN, RNN, and Transformer"
_ICLR.cc/2022/Conference — ICLR 2022 Poster_

### Official Review · Reviewer_pfre · 2021-11-02

**Correctness:** 3
**Technical Novelty And Significance:** 3
**Empirical Novelty And Significance:** 3
**Recommendation:** 5
**Confidence:** 4

**Main Review:**

First, as in Figure 2, the paper proposes to map input x and output y into a latent space for metric learning ($f(x)=g(y)$) and auto-encoder learning ($y=h(g(y))$) are also investigated in multi-label classification[r1,r2], which are not discussed in this paper. In my opinion, the main difference is the design of multiple latent spaces compared with these multi-label classification methods.


Second, in traditional machine learning, we often map a high dimensional space to a low dimensional space for metric learning. It is unclear why maps the target $y$ to the intermediate layers in this paper. Given a high dimensional space (e.g., images), the inference model extracts useful features and filters unrelated features for metric learning.

However, in this paper, I find that the authors conduct experiments on some single label classification (e.g., CIFAR10 and Fashion-MNIST). In this case, $y$ is a scalar or one-hot vector, I am curious about the exact form of $g_1$, $g_2$, $g_3$ in Figure 2. Does the proposed method map a low dimensional latent space to a high one? What is the motivation for expanding representation space? If $g_1(y)$ and $g_2g_1(x)$ are still in a low dimensional space or $g_i$ are very simple, do we really need inverse transformations from $Y$ to AL layers? In this case, we can simply fuse different AL layers to top layers for metric learning. For example, we can move $y$ after  $t_3$ and remove $h_1$, $h_2$, $h_3$ in Figure 2. Since $y$ is a specific label, it is unclear why we need to map to a high dimensional space.

The design of multi-label classification is reasonable to me because the target $y$ is complex (e.g., the multiple label vectors could miss some labels) and the multi-labels could be in a high dimensional space. In this case, one can map high dimensional space $X$ and $Y$ into a low dimensional latent space for metric learning.

[r1] Learning Deep Latent Spaces for Multi-Label Classification, AAAI 2017.
[r2] Multi-label Classification via Feature-aware Implicit Label Space Encoding, 2014.
...

Third, it would be better to set a baseline by moving $y$ after  $t_3$ and removing $h_1$, $h_2$, $h_3$ in Figure 2 for comparison.

Fourth, the architecture of AL layers is similar to Ladder networks. It is suggested to analyze the differences.

[r3] Semi-Supervised Learning with Ladder Networks, 2015.

**Summary Of The Paper:**

This paper proposes associated learning (AL) for CNN, RNN, and transformer. Different from back-propagation (BP), AL decomposes BP’s global end-to-end training strategy into several small local optimization targets such that each sub-networks has an isolated gradient flow. To achieve this, the paper proposes to map input $x$ and output $y$ into intermediate AL layers and performs metric learning (e.g., $t_1=b_1(s_1)$) and auto-encoder learning ($t_1=t_1^{‘}$), as shown in Figure 2. Moreover, Each AL layer can be optimized locally. The idea is interesting. The experiments demonstrate the effectiveness on (IMDB Review, AG’s News corpus, DBpedia Ontology, the Stanford Sentiment Treebank, CIFAR10, and Fashion-MNIST.

**Summary Of The Review:**

(1) The proposed method can be optimized locally, and achieve competitive results. The proposed framework can be used for CNNs, RNNs, and transformers. The idea is interesting.

(2) More analyses about the motivation and the necessity of inverse transformation for Y to latent space are needed.

(3) The analyses and discussions about related works, such as multi-label classification and ladder networks, are missed.

(4) Some experiments are suggested to support the authors' opinions (e.g. (2) and (3)) if possible.

---

> ### Author Response · Authors · 2021-11-15
> **We thank the reviewer for reviewer's thoughtful and thorough review**
>
> **Q1 Analyses about the motivation and the necessity of inverse transformation for Y to latent space:**
>
> Thanks for this insightful comment. Indeed, mapping a one-hot encoded label into a latent representation may seem counter-intuitive, as the reviewer pointed out. However, we experimented with the structure suggested by the reviewer (e.g., move $y$ after $t_3$ and remove $h_1$, $h_2$, and $h_3$ in Figure 2) on the AGNews dataset using the LSTM network and the Transformer (denoted as "LSTM-AL w/o AE" and "Tran-AL w/o AE," respectively). The results show that our proposed AL model (mapping $y$ to a latent space and transforming it back to $y$ using an autoencoder) makes better predictions on the test data. The following two tables give the result.
>
> For both LSTM and Transformer network structures, training based on associated learning is better than end-to-end backpropagation, which is better than the associated learning without autoencoders. We suspect that each autoencoder perhaps performs some kind of feature extraction and regularization. Particularly, although it is easy to transform one-hot encoded $y$ to $t_1$ to decrease associated loss and transform $t_1$ to  $t_0^{\prime}$  to decrease the autoencoder loss, the network still needs to learn more challenging task: converting $t_i$ to $t_{i+1}$ and $t_{i+1}$ to $t'_i$ for $i > 1$ to decrease the associated loss and autoencoder loss of layer $i$.  Consequently, adding autoencoders may prevent overfitting for the components beyond layer 1.
>
> Additionally, we experimented with the reviewer's proposed architecture based on a one-layer Transformer and a two-layer Transformer, denoted as "Tran-AL w/o AE (2 layers)" and "Tran-AL w/o AE (1 layer)", respectively. We found that adding more layers may harm the prediction power (on average, adding the second layer reduces the accuracy from 89.08% to 88.31%). Thus, although the reviewer's proposed architecture looks promising, initial experiments show that the reviewer's method may still perform unsatisfactorily. We will include this experiment in our ablation study section.
>
> **Additional ablation experiment by removing reversed auto encoders**
>
> LSTM on AGNews
>
> | Model | Test Accuracy|
> | ----------- | ----------- |
> | LSTM(2-layers) | $90.32 \pm 0.23$  |
> | LSTM-AL (2 layers) | **$91.53 \pm 0.20$** |
> | LSTM-AL w/o AE (2 layers)| $89.39 \pm 0.99$ |
> | LSTM-AL w/o AE (1 layer)| $89.30 \pm 0.88$ |
>
> ____
> ______
>
> Transformer on AGNews
>
> | Model | Test Accuracy|
> | ----------- | ----------- |
> | Tran (2 layers) | $90.71 \pm 0.28$  |
> | Tran-AL (2 layers) | **$91.17 \pm 0.43$** |
> | Tran-AL w/o AE (2 layers) | $88.31 \pm 0.55$ |
> | Tran-AL w/o AE (1 layer)| $89.10 \pm 0.71$ |
>
> ___
>
>
>
> **Q2 Analyses and discussions about related works, such as multi-label classification and ladder networks**
>
> Thanks for pointing out these relevant works.  We carefully reviewed these works and compared these methods with AL in the paper.
>
> Indeed [r1, r2] also transforms $x$ into latent space, which is further transformed into $y$ via an autoencoder.  Although this design is structurally similar to AL, their design motivations are different.  These works attempt to convert a multi-label target $y$ into a low-dimensional latent representation $c$, so learning a function to convert $x$ to $c$ may require fewer training samples.  Our target is to design isolated loss functions so that the new design has favorable properties that BP does not have, such as short gradient flow, forward shortcuts, dynamic layer accumulation, and layerwise pipeline learning.  Our method to regionalize local loss functions happened to be structurally similar to [r1, r2].
>
> The ladder network [r3] uses a denoising autoencoder to extract the essential features, used as the input to predict targets.  In addition, the ladder network adds skip connections between each encoder to its corresponding decoder.  While the ladder network may look similar to AL structurally, at least two manifest differences exist.  First, the residual between $y$ and $\hat{y}$ in the ladder network is propagated through all layers to update the parameters on the inference path, but in AL, each layer has a local objective function, and most layers do not receive signals from the output layer.  Second, the skip connection in the ladder network connects an encoder layer to a decoder layer. In contrast, the bridge function in AL connects a hidden representation of $x$ to the bottleneck layer of an autoencoder to create the local loss.

---

> > ### Author Response · Authors · 2021-11-25
> > **Looking forward to hearing from you**
> >
> > Dear reviewer pfre,
> >
> > We want to send you a friendly reminder for the discussion. Here is a summary of our response for your valuable feedback!
> >
> > * We conducted a new ablation study about the necessity to map $y$ into latent space.
> > * We discussed the ladder networks in **section 4.2**
> >
> > We thank you again for your valuable comments, and we would appreciate it if you could reconsider the evaluation of our work based on our response. We are also happy to extend our response if you have any other concerns.
> >
> > Thanks.

---

> > > ### Comment · Reviewer_pfre · 2021-11-25
> > > **The idea of mapping one-hot vectors into a high dimensional space**
> > >
> > > Thanks for the response to my questions. However, the main concern about the idea that mapping one-hot vectors into a high dimensional space is not clear for me. I have read the PyTorch pseudocode of a 5-layer AL network at the end of the paper. I find that $g_1$ is a MLP, whose size is $(class_num,10)$. Take FashionMNIST as an example, if an example ``5" is fed into the network, its one-hot vector is $(0,0,0,0,1,0,0,0,0,0)$. The proposed method projects it into a 10-dim embedded feature. I do not know why we need such a kind of projection here. The author did not provide an in-depth theoretical analysis of this point, as discussed above.  I concern that experiments could be only effective on toy networks. For example, in Figure 4, the CIFAR-10 only obtains 85\% accuracy while SOTA networks obtained 95\% accuracy several years ago.

---

> > > > ### Author Response · Authors · 2021-11-28
> > > > **Response to reviewer pfre**
> > > >
> > > > Thank you for the response.
> > > >
> > > > **Q1: Why map one-hot vectors into a high dimensional space?**
> > > >
> > > >
> > > > Mapping one-hot encoded $y$ to a high dimension space can accelerate the convergence speed for the reasons explained below.
> > > >
> > > > AL can be regarded as a particular form of supervised contrastive learning (SCL) such that every sample’s latent representation is close to the embedding of a class.  According to [r1], when using SCL for classification, the cross-entropy loss attains its minimum if and only if the features of each class collapse to the vertices of an origin-centered regular K-1 simplex(K = class_num) (Figure 3 in [r1]). Since one-hot encoded $y$ is a K-1 simplex, $g_1(y)$ would also be a K-1 simplex. Since we obtain such K-1 regular simplex in the early stage of training (i.e., letting $b_1(f_1(x)) \approx g_1(y)$), the associated loss directly optimizes for a K-1 regular simplex in the representation space, which is the best configuration of minimum SCL loss. In other words, such an autoencoder setup helps AL find the best configuration directly, and the bridge functions help to find the best configuration early.  This insight explains why AL converges so fast and why we map $y$ into latent space.
> > > >
> > > > Empirical studies (shown in the previous reply) also demonstrate that removing all autoencoders and using $b_i$ as the classifier to predict $y$ is not helpful, and adding the autoencoder improves the test accuracy.
> > > >
> > > > We will add this discussion in our paper.
> > > >
> > > > [r1] Graf, Florian, et al. "Dissecting supervised constrastive learning." ICML 2021.
> > > >
> > > > **Q2: Perhaps AL is effective only on toy networks?**
> > > >
> > > > The purpose of this paper is to propose an alternative to end-to-end backpropagation (since BP has well-known problems), not to propose a new architecture to beat SOTA on an experimental dataset. Additionally, to obtain a SOTA performance, we usually need various preprocessing steps (e.g., data augmentation, pretraining), which is beyond the scope of our study. Therefore, we compared AL and BP on some of the most famous network architectures, including toy networks (such as vanilla CNN) and more complex networks (such as LSTM, Transformer, and VGG).
> > > >
> > > > For a fair comparison, we used the settings and hyperparameters reported in the papers and the implementations on GitHub. We believe our results of BP are comparable to the SOTA results under the same or similar network structures. Some examples are listed below.
> > > >
> > > > (1) The original VGG paper conducted experiments on neither CIFAR-10 nor fashion-mnist . However, according to https://github.com/chengyangfu/pytorch-vgg-cifar10, the accuracy of VGG-16 on CIFAR-10 is 92.63%, and our VGG-16 (BP) on CIFAR-10 is 92.14%.  Additionally, according to https://github.com/zalandoresearch/fashion-mnist, the accuracy of VGG-16 on fashion-mnist is 93.5%, and our VGG-16 (BP) on fashion-mnist is 94.18%.
> > > >
> > > > (2) According to “Baseline Needs More Love: On Simple Word-Embedding-Based Models and Associated Polling Mechanisms in ACL2018”, the accuracies of LSTM on AGNews and DBpedia are 86.06% and 98.55, respectively, and our LSTM (BP) results are 90.32% and 97.34, respectively.(MLP classifier dimension, vocabulary construction and other hyperparameters might be different)

---

### Official Review · Reviewer_L9xu · 2021-11-04

**Correctness:** 3
**Technical Novelty And Significance:** 3
**Empirical Novelty And Significance:** 3
**Recommendation:** 5
**Confidence:** 3

**Main Review:**

Thank you for this read. The results and the methodology are definitely compelling. Why I cannot accept the manuscript as is, is that:
· The motivation is not clear enough. It is clear wrt why BP is not ideal. But it is not clear how you landed on this method specifically, as compared to many other attempts on finding more optimal neural network optimization methodologies.
· Section 2 is very difficult to follow. I would spend some more effort explaining how your method works in the manuscript.
· It would be nice to include an algorithm of how to implement AL.

A selection of minor comments:
· Some typos throughout the manuscript, e.g., in abstract "associate" and paragraph 4 in the introduction "in Section 4 We".
· Notation must be introduced, e.g., f, y, etc. in Section 2 are not introduced properly in relation to Figure 1.
· It is difficult to follow the difference in notation when using h, b, and f. I recommend you spend some more time on making this very clear to the reader.
· I find the Table 2 epochs for AG News difficult to follow. There is a clear pattern that AL is faster, but then things changes radically for AG News? Would be nice with some further analysis into this.

**Summary Of The Paper:**

This paper studies and benchmarks an alternative to back-prop named associated learning. They analyze the pros and cons.

**Summary Of The Review:**

With some more clarification on how you ended up with this methodology and a clear algorithm for how to implement AL, the reviewer would be happy to accept the manuscript.

---

> ### Author Response · Authors · 2021-11-15
> **Thank you for your work in reviewing our paper and thoughtful advices!**
>
> **Q1 Give some more clarification on how you ended up with this methodology:**
>
> We want to decompose the end-to-end backpropagation into small isolated components such that each component has a local objective.  Our design is biologically plausible because each gradient flow is short, and only neighboring neurons can exchange information.  On the contrary, previous studies on BP alternatives with separated gradient flows usually pass the target or a global loss into each local component, which could be biologically implausible because signals are unlikely to connect to several neurons that are far from each other.  Additionally, our design may bring favorable properties that BP does not have, as mentioned in Section 2.2.
>
> To localize objective function, our first intuition is to convert the input $x$ and output $y$ into latent representations $s_i$ and $t_i$ for each layer $i$ and use the distance between $s_i$ and $t_i$ as the local objective function for layer $i$.  This idea inspires us to design the forward functions $f_i$ and $g_i$ for every layer (referring to Figure 2).  In order to compare $s_i$ and $t_i$, we add a function $b_i$ to convert $s_i$ into $s’_i$ such that the shapes of $s’_i$ and $t_i$ are identical, so they are comparable.  Although the design so far allows converting both $x$ and $y$ into a latent space, transforming from a latent representation into $y$ at the inference phase is still an issue.  Eventually, we attach an autoencoder in each layer: the above mentioned function $g_i$ can be considered as an encoder to encode $y$ into latent space, and we add a decoder $h_i$ such that $h_i(g_i(t_i)) = t’_i \approx t_i$.
>
> **Q2 A clear algorithm for how to implement AL:**
>
> Algorithms were added in the new version of our paper (please refer to **section 2.1**  and **appendix  6.1**)
>
> **Q3 It is difficult to follow epochs for AG News in Table 2:**
>
> Training an LSTM, a CNN, or a VGG based on AL requires much fewer epochs when compared with training them by BP. However, this trend is less clear when training a Transformer-AL model.  We carefully reproduce Transformer-AL on AGNews. We found that Transformer-AL reaches high validation and test accuracy within the first 5 epochs, but the best validation and test accuracy come at ~20 epochs. Since we report only the epoch that reaches the best validation and test accuracies, the required epochs look large.  Experiments of Transformer-AL on DBpedia show similar results.
>
> Details: after training the Transformer-AL on AGNews for 5 epochs, the validation accuracy (90.95) can already achieve 99% of the best validation accuracy (91.31), and the test accuracy (90.98) is very close to the best test accuracy (91.17, i.e., training Transformer-AL for ~20 epochs for AG News).  After training the Transformer-AL on DBpedia for 5 epochs, its validation accuracy achieves 96.0902, which is very close to the best validation accuracy 96.6134.

---

> > ### Author Response · Authors · 2021-11-25
> > **Looking forward to hearing from you**
> >
> > Dear reviewer L9xu,
> >
> > We want to send you a friendly reminder for the discussion. Here is a summary of our response for your valuable feedback!
> >
> > * We rewrite the section 2 with algorithms, making it easier to follow
> > * The motivation of AL can be found in **section4.1**
> > * We added the discussion and analysis of Transformer converge epoch in **section6.6**
> >
> > We thank you again for your valuable comments, and we would appreciate it if you could reconsider the evaluation of our work based on our response. We are also happy to extend our response if you have any other concerns.
> >
> > Thanks.

---

### Official Review · Reviewer_dmxT · 2021-11-05

**Correctness:** 4
**Technical Novelty And Significance:** 2
**Empirical Novelty And Significance:** 2
**Recommendation:** 6
**Confidence:** 4

**Main Review:**

The paper clearly lays out the advantages of associated learning: faster inference, dynamic layer accumulation, and pipeline. The paper is clearly written with good figures. The experiments appear to be easy reproducible, too. The decrease in epochs needed for LSTMs is particularly impressive.

I found the biological basis a little lacking. Perhaps, some type of curriculum learning or more exploration on what the various shortcuts are doing could make this argument stronger. The related works section neglects to mention other gradient-isolated methods like https://arxiv.org/abs/1905.11786. I think in some ways this work can be seen as encoder-decoder with additional regularization, too?

**Summary Of The Paper:**

Associated Learning puts forth a template that can be applied to almost any network to achieve faster training and inference. They apply their template to several existing deep learning models and perform experiments that show they can achieve comparable if not better results with less training time.

**Summary Of The Review:**

I would recommend this paper to be accepted. While there are several issues, the empirical results are strong (particularly the LSTM reduction in epochs). I think is a lot more to explore with the dynamic layer accumulation and gradient isolation, too, that would be interesting to other researchers.

---

> ### Author Response · Authors · 2021-11-15
> **Thank you for your great review and suggestions!**
>
> **Q1 The biological basis is a little lacking:**
>
> Biological plausibility can be interpreted from different perspectives.  AL is biologically plausible because each neuron only exchanges information with neighboring neurons within the same AL layer.  On the other hand, previous BP alternatives sometimes need to connect a neuron to several neurons that may be far away, e.g., Nøkland & Eidnes, 2019.  Additionally, each AL layer has a local objective function rather than a global objective function; a global objective function is unlikely to exist in the brain.  That being said, we admit that the primary purpose of the paper is not to propose a biologically plausible model. Instead, our motivation is to decompose end-to-end backpropagation into isolated components such that each layer has a local objective function, which leads to favorable properties that end-to-end BP does not have, as explained in Section 2.2.
>
> **Q2 Some type of curriculum learning or more exploration on what the various shortcuts are doing could make this argument stronger:**
>
> Thanks for the great suggestion.  Although we stop the gradient flow between different AL layers, the latent representations in the lower layer appear to help generate the latent representations in an upper layer.  Details are described below.
>
> We experimented with the AL layer number $i$ vs. $|s’_i - t_i|_2^2$ using different datasets on different network structures.  The following tables show that the associated loss at an upper layer (i.e., the layers closer to the output) is smaller than at a lower layer (i.e., the layers closer to the input).  Consequently, the shortcuts are likely doing some kind of curriculum learning, as the reviewer suggested.
>
> We will put the new result in the Appendix.
>
> **DBpedia LSTM-AL**
>
> | Layer      | Associated Loss|
> | ----------- | ----------- |
> | Embedding layer      | $5.6061 \times 10^{-5}$       |
> | LSTM-1  | $3.7440 \times 10^{-5}$        |
> | LSTM-2   | $1.7786 \times 10^{-7}$        |
>
> ___
>
> **AGNews LSTM-AL**
>
> | Layer      | Associated Loss|
> | ----------- | ----------- |
> | Embedding layer      | $1.4802 \times 10^{-4}$       |
> | LSTM-1  |$5.3281 \times 10^{-5}$  |
> | LSTM-2   | $4.8651 \times 10^{-6}$    |
>
> ___
>
> **AGNews Transformer-AL**
>
> | Layer      | Associated Loss|
> | ----------- | ----------- |
> | Embedding layer      | $3.31787 \times 10^{-4}$     |
> | Layer-1  |$1.2561 \times 10^{-4}$  |
> | Layer-2   | $1.3976 \times 10^{-5}$   |
>
> ___
>
> **DBpedia Transformer-AL**
>
> | Layer      | Associated Loss|
> | ----------- | ----------- |
> | Embedding layer      | $5.8318 \times 10^{-5}$    |
> | Layer-1  |$3.0053 \times 10^{-6}$ |
> | Layer-2   | $7.4727 \times 10^{-8}$   |
>
> ___
>
>
> **Q3 Comparison with a related paper “Putting An End to End-to-End: Gradient-Isolated Learning of Representations”:**
>
> Thanks for pointing out this interesting related work.  We will explain the relationship between this related work and our paper in Section 4.
>
> This related work proposed Greedy InfoMax (GIM), which indeed shares similarities with our AL: both models introduce local (isolated) objective functions for each layer.  However, they still differ in some ways.  In GIM, the local loss function is constructed based on contrastive loss.  Consequently, GIM can mainly be applied to self-supervised tasks.  On the other hand, our AL designs each local objective function by the distance between latent representations of input $x$ and output $y$.  As a result, for supervised learning tasks, AL is likely a more natural choice than GIM.

---

> > ### Author Response · Authors · 2021-11-25
> > **Looking forward to hearing from you**
> >
> > Dear reviewer dmxT,
> >
> > We want to send you a friendly reminder for the discussion. Here is a summary of our response for your valuable feedback!
> >
> > * We explained the biological motivation from the angle for signal passing (can be found in **section 4.2**)
> > * We discussed GIM and AL
> > * We added experiment results related to curriculum learning (can be found in **Appendix 6.7**)
> >
> > We thank you again for your valuable comments, and we would appreciate it if you could reconsider the evaluation of our work based on our response. We are also happy to extend our response if you have any other concerns.
> >
> > Thanks.

---

### Official Review · Reviewer_9Led · 2021-11-09

**Correctness:** 3
**Technical Novelty And Significance:** 2
**Empirical Novelty And Significance:** 2
**Recommendation:** 6
**Confidence:** 3

**Main Review:**

The authors have resolved my concerns on the technical details of how AL is applied on RNNs and Transformers.

================================

The paper is well written. Experiments show that in text classification and image classification, the proposed method outperform BP in some basic architecture setting.

Here are my concerns:

- It is unclear how the AL is applied on RNNs and Transformers. Section 2.1.1 only very briefly described them, but I could not figure out some of the details. For example, how is the temporal data processed in LSTM?

-In CNN, when flatting the hidden representation, it also lost the spatial information in feature maps. Furthermore, how to convert si to si' if ti is also a 3d feature map when the spatial information is lost.

- From the description in Section 2, it seems that AL introduces around double the parameters for a given neural network. What is the impact of the increased parameters in computation cost?

- The experiments uses relatively simple network architecture for text classification. Does the same benefits carry over to large transformer models, and still beats currently popular models like BERT?

- The architecture information on CNN in section 3.3 is missing.

- If my understanding is correct, the proposed architecture would not work in sequence generation task like LSTM and transformers could do. Right?

**Summary Of The Paper:**

This paper is a continuation of an original associated learning paper by Kao&Chen 2021. It attempts to propoose new learning approach associated learning as an alternative way to back-propagation. On top of the original paper, it discovers more interesting properties and extend AL to CNN, LSTM and transformers (though lacking sufficient details).

**Summary Of The Review:**

In summary, I think though the paper proposes AL framework as an alternative to BP, it is actually a simple extension to a previous work, and does not proposes substantially new ideas. Some details are missing, and experiments are not extensive enough to cover state-of-the-art architectures.

---

> ### Author Response · Authors · 2021-11-15
> **Thank you for your insightful review and advices!**
>
> **Q1 Experiments are not extensive enough to cover state-of-the-art architectures, e.g., BERT.**
>
> We admit that AL is not applied to state-of-the-art architectures.  However, previous studies mostly show the effectiveness of their proposed BP alternatives on few selected networks.  To the best of our knowledge, this is the first paper that experiments with BP alternatives on a wide range of representative neural network structures (including vanilla CNN, VGG, LSTM, Bi-LSTM, and Transformer) and obtains decent results in most cases.  We believe that this is a considerable improvement compared to the previous studies on BP alternatives.
>
> **Q2 How to apply AL on RNN and Transformer when the input feature is sequential:**
>
> AL can be applied to various network structures on various supervised learning tasks with slight modifications on the associated loss. We use text sequence as the input sequence for the following explanation.
>
> A typical RNN iteratively transforms an input word token and the previous hidden state into a new hidden state.  When converting an RNN into its AL form, the RNN-AL processes the input text sequence almost the same as a typical RNN: the RNN-AL also iteratively transforms an input token and the hidden state into a new hidden state.  After reading the entire input tokens, we define the latent representation of the input sequence $x$ as the final hidden state.  Consequently, the associated loss is defined as the distance between the final hidden state and the latent representation of $y$.  In this manner, we may apply AL on RNN with sequential input features.
>
> The Transformer-AL encodes the input sequence $x$ as a regular Transformer does. We define the latent representation of $x$ by computing the mean-pooling on the encoded token vectors. Therefore, the associated loss is defined as the distance between the latent representations of $x$ and $y$.
>
> **Q3 How to apply AL on CNN? Spatial information is lost when flatting the hidden representation:**
>
> The CNN-AL transforms an input image $x$ into a latent representation by convolutions, just like a regular CNN does.  The output feature map is regarded as the latent representation of $x$.  Therefore, the spatial information of $x$ is preserved.  The associated loss is defined as the distance between the flattened feature map and the latent representation of $y$.
>
> **Q4 The proposed architecture would not work in sequence generation task:**
>
> AL can be applied to sequence generation tasks.  Experiments on this type of task are left as future work.  Below, we use machine translation as an example to explain the sequence-to-sequence model using AL.
>
> The input text sequence $x$ is transformed into a latent representation based on LSTM or Transformer using the methodology we introduced in Q2.  The output text sequence $y$ is converted into a latent representation in the same manner.  The associated loss is defined as the distance between the latent representations of $x$ and $y$.  At the inference stage, we could use customized autoencoders (e.g., LSTM-autoencoder [r1]) to generate the word tokens one by one in the target language.
>
> [r1] Chung, Yu-An, et al. "Audio word2vec: Unsupervised learning of audio segment representations using sequence-to-sequence autoencoder." arXiv preprint arXiv:1603.00982 (2016).
>
> **Q5 The impact of the increased parameters in computation cost:**
>
> For a fair comparison between AL and BP, we ensure the number of parameters in AL to be close to (but not larger than) the number of parameters in BP for all the comparisons reported in our paper.  However, AL indeed requires more training time per epoch probably because PyTorch and Tensorflow need to process multiple computational graphs for weight update in AL than in BP. In our current setting, the ratio of the training periods of one epoch in AL and BP is approximately 1:0.7. However, as shown in Table 2, Table 3, and Table 4, AL usually requires far fewer epochs to obtain decent results. Moreover, AL can pipeline the training such that the parameters in different layers are updated simultaneously, which may further increase the training throughput.

---

> > ### Author Response · Authors · 2021-11-25
> > **Looking forward to hearing from you**
> >
> > Dear reviewer 9Led,
> >
> > We want to send you a friendly reminder for the discussion. Here is a summary of our response for your valuable feedback!
> >
> > * Responses for more state-of-the-art models
> > * We explained how AL applied on CNN and RNN
> > * We showed possible AL design for sequence generation tasks
> > * We controlled the parameter amount between AL and BP, indeed, AL requires more time to train an epoch under the experiment environment we used
> >
> >
> > We thank you again for your valuable comments, and we would appreciate it if you could reconsider the evaluation of our work based on our response. We are also happy to extend our response if you have any other concerns.
> >
> > Thanks.

---

> > > ### Comment · Reviewer_9Led · 2021-11-25
> > > **Details on the associated loss in RNN and Transformer**
> > >
> > > Thanks for the response. It is clear that The associated loss is defined as the distance between the flattened feature map and the latent representation of y. But what about the intermediate layer loss between si' and ti'? I have the same question as Reviewer pfre. If for example, two layer RNN, there should be s1, s2, right? What is the associated loss applied on the first layer RNN output? How does the label y be transformed such that there are also t1, t2? This part is not clear to me.

---

> > > > ### Author Response · Authors · 2021-11-26
> > > > **Response to reviewer 9Led**
> > > >
> > > > Thank you for the response.
> > > >
> > > > For CNN-AL, the associated loss is defined as the distance between the flattened feature map and the latent representation of $y$.  For RNN-AL, we use the last token’s latent state as the latent representation of the entire input sequence $x$, and compute associated loss with $t$. For Transformer-AL, we use mean-pooling on the encoder's output vector as the latent representation of the input sequence $x$.  Details are described below.
> > > >
> > > > Figure 1 shows a unfold RNN-AL example
> > > > [Figure1](https://i.imgur.com/o2mEDKC.jpeg)
> > > >
> > > > Figure1 (above link) shows the AL form of a 2-layer RNN and the unfolded network.  Each token $x^{(i)}$ in the input sequence $x = [x^{(1)}, x^{(2)}, \ldots, x^{(|x|)}]$ is fed into RNN-AL sequentially.  Given an token $x^{(i)}$ and previous hidden state $s_1^{(i-1)}$, RNN-AL generates the new latent state $s_1^{(i)}$, which represents the hidden state after processing the first $i$ tokens.  We use the $s_1^{(|x|)}$ the last hidden state as the latent representation of $x$ at the first layer, since $s_1^{(|x|)}$ is the hidden state after processing the entire $x = [x^{(1)}, x^{(2)}, \ldots, x^{(|x|)}]$.  We generate the latent representation of $x$ in other layers in the same manner.
> > > > _____
> > > > Figure 2 shows a unfold Transformer-AL example
> > > > [Figure2](https://i.imgur.com/e1MfMb8.jpeg)
> > > >
> > > > Figure2 shows the Transformer-AL and the unfolded network.  The input sequence $x = [x^{(1)}, x^{(2)}, \ldots, x^{(|x|)}]$ is fed into an attention layer to generate $[s_1^{(1)}, \ldots, s_1^{(|x|)}]$.  Transformer-AL applies mean-pooling on the output vector to obtain $s_1$ the latent representation of $x$ in the first layer.  We can process the latent representation of $xt at other layers in the same manner.
> > > >
> > > > As for the latent representation of $y$, we use a MLP $g_1$ to map $y$ into a latent representation $t_1$, which should be close to $s'_1$.  We use another MLP $h_1$ to map $t_1$ back to $y$.
> > > >
> > > > We hope this explanation can answer your concerns.  We will add these explanations and Figures into Appendix.

---

### Author Response · Authors · 2021-11-15
**General Response to all reviewers**

End-to-end backpropagation (BP) is a cornerstone of today’s deep learning.  However, it still has several problems, so attempts to replace end-to-end backpropagation deserve broad discussions in ICLR.

To the best of our knowledge, this is the first paper to apply BP alternatives to a wide range of network structures, including the family members in MLP, CNN, RNN, and Transformer, and obtain comparable (if not better) results to BP in most cases.  As a result, this study may encourage more researchers to study BP alternatives.

Since BP is the default learning strategy in today’s deep learning, current best practices (e.g., dropout, batch norm, and activation function) are tailored for BP.  Even though these settings may not necessarily be the best settings for AL, AL still yields comparable performance to BP.  We would like to introduce AL to the public so that the researchers can experiment with different settings and perhaps discover the best settings for AL to maximize its effectiveness.

---

### Author Response · Authors · 2021-11-21
**Manuscript updated**

Dear reviewers,

Thank you for your great suggestions. We have modified our manuscript as suggested.

The motivation behind AL can be found in the discussion (**Section 4.1**)

1. We introduced algorithms for constructing and training AL networks (**Section 2.1**). In addition, we added the inference algorithm (**Appendix 6.1**).
2. We experimented with removing all autoencoders and using $b_i$ as the classifier to predict $y$ (similar to the early exit technique) (**Section 3.5**).
3. We discussed more related works (**Section 4.2**).
4. We reported experimental details (**Appendix 6.2 - 6.5**)
5. We reported details on the convergence speed of the Transformer (**Appendix 6.6**).
6. We conducted experiments related to curriculum learning (**Appendix 6.7**).
7. We showed PyTorch pseudocode (**Section 6.8**).

---

### Decision · Program_Chairs · 2022-01-20

**Decision:**

Accept (Poster)

**Comment:**

The authors propose a method for associative learning as an alternative to back propagation based learning. The idea is to interesting. The coupling between layers are broken down into local loss functions that can be updated independently. The targets are projected to previous layers and the information is preserved using an auto-encoder loss function. The projections from the target side are then compared with the projections from input side using a bridge function and a metric loss. The method is evaluated on text and image classification tasks. The results suggest that this is a promising alternative to back propagation based learning.

Pros
+ A novel idea that seems promising
+ Evaluated on text and image classification tasks and demonstrated utility

Cons
- The impact of the number of additional parameters and the computation is not clarified (even though epoch's are lower)

The authors utilized the discussion period very well, running additional experiments that were suggested (especially ablation studies). They  also clarified all the questions that were raised. In all, the paper has improved substantially from the robust discussion.